# Postdiction in Visual Awareness in Schizophrenia

**DOI:** 10.3390/bs12060198

**Published:** 2022-06-20

**Authors:** Szabolcs Kéri

**Affiliations:** 1Department of Cognitive Science, Budapest University of Technology and Economics, H1111 Budapest, Hungary; keri.szabolcs@ttk.bme.hu; 2Nyírő Gyula National Institute of Psychiatry and Addictions, H1135 Budapest, Hungary; 3Hungarian Association for Behavioral, Cognitive, and Schema Therapy, H1083 Budapest, Hungary; 4Department of Physiology, Albert Szent-Györgyi Medical School, University of Szeged, 6720 Szeged, Hungary

**Keywords:** schizophrenia, cognition, postdiction, perception, cognitive-behavioral therapy

## Abstract

Background: The mistiming of predictive thought and real perception leads to postdiction in awareness. Individuals with high delusive thinking confuse prediction and perception, which results in impaired reality testing. The present observational study investigated how antipsychotic medications and cognitive-behavioral therapy (CBT) modulate postdiction in schizophrenia. We hypothesized that treatment reduces postdiction, especially when antipsychotics and CBT are combined. Methods: We enrolled patients with schizophrenia treated in a natural clinical setting and not in a randomized controlled trial. We followed up two schizophrenia groups matched for age, sex, education, and illness duration: patients on antipsychotics (*n* = 25) or antipsychotics plus CBT (*n* = 25). The treating clinician assigned the patients to the two groups. Participants completed a postdiction and a temporal discrimination task at weeks 0 and 12. Results: At week 0, postdiction was enhanced in patients relative to controls at a short prediction–perception time interval, which correlated with PANSS positive symptoms and delusional conviction. At week 12, postdiction was reduced in schizophrenia, especially when they received antipsychotics plus CBT. Patients with schizophrenia were also impaired on the temporal discrimination task, which did not change during the treatment. During the 12-week observational period, all PANSS scores were significantly reduced in both clinical groups, but the positive symptoms and emotional distress exhibited a more pronounced response in the antipsychotics plus CBT group. Conclusion: Perceptual postdiction is a putative neurocognitive marker of delusive thinking. Combined treatment with antipsychotics and CBT significantly ameliorates abnormally elevated postdiction in schizophrenia.

## 1. Introduction

Despite its common occurrence and clinical significance, a long-standing debate persists about how we can conceptualize delusions and how skewed neurocognitive mechanisms contribute to “a false belief based on incorrect inference about external reality” [1]. Cultural acceptance and adaptive social features play a pivotal role in differentiating pathological and normal beliefs: delusions are socially maladaptive and “not ordinarily accepted by other members of the person’s culture or subculture” [1]. Although the American Psychiatric Association’s statement seems to be clear-cut, evidence from clinical psychology, cultural anthropology, and cognitive science suggests a continuum between normal belief formation and psychopathological phenomena, with a blurred boundary between the community’s norms and disorders of thought that require clinical attention [2,3,4].

Many patients with schizophrenia exhibit neurocognitive alterations, including attention, executive functions, memory, and sensory information processing [5,6,7]. In addition, many domains of social cognition are also compromised in psychotic disorders, resulting in difficulties in experiencing and recognizing emotions, motivation, and attributing mental states to other people [8,9,10]. However, it is not clear how these neurocognitive and social cognitive abnormalities lead to complex alterations in the internal modeling and representation of the world in psychoses.

Numerous approaches that delve into the mechanism of biases in belief formation focus on delusions as models, including abnormal attribution processes, inferential reasoning, belief evaluation, metacognition, error-dependent updating, preconscious perceptual processing, and belief–memory interferences [11,12,13,14,15,16,17]. In Fleminger’s (1992) model [18], beliefs and expectations fundamentally influence the interpretation of perceptual information. Expectations can be implicit and unconscious (“self-reinforcing preconscious cycles of perceptual processing”), or, in other cases, explicit and conscious beliefs may change the nature of perceptual experiences (“we see what we believe”). In other words, individuals may perceive the world as if their beliefs were true without active inference and adaptive adjustments of representations [19].

Sometimes thoughts on expectations and predictions can be confused with percepts. Namely, mistiming perception and thought leads to postdiction in awareness, which may be a critical factor in delusions [20,21,22,23,24]. According to Bear et al. (2017), the following example illustrates postdiction: “Imagine that, as you leave your house, a few raindrops fall on your skin. You may have the thought that you should go grab your umbrella. Such an observation is completely ordinary and unlikely to encourage any odd beliefs about how the world works. However, a minor alteration to the order in which this perception and thought arise might produce a dramatically different outcome. Mistakenly thinking that you knew to grab your umbrella before you felt raindrops might inspire the belief that you have an exceptional ability to predict the weather or even that you are clairvoyant. More generally, someone who systematically misperceives herself as successfully predicting an event like the weather could come to hold exaggerated or even delusional beliefs about her knowledge or agency”.

The sense of agency, linking intentions to actions, is essential to understanding psychotic symptoms, especially in the case of some types of delusions (e.g., thought broadcasting, passivity phenomena, and the sense of being controlled by external forces). Patients with schizophrenia are often impaired in predicting the consequences of willed actions and under- or over-attribute agency to themselves [25,26]. To establish a sense of agency, humans use both prospective (predictive) and retrospective (outcome-dependent and inferential) processes in close interaction [27]. Di Plinio et al. (2020) [28] showed that an ongoing event retrospectively influenced prospective intentional attribution during a short temporal window, which is similar to postdiction. This retrospective–prospective interaction was weak in individuals with high psychotic-like experiences [28].

Notably, a simple perceptual postdiction task can precisely tap into the formation of overvalued ideas and delusions [21]. During the task, participants see five empty squares on a computer display and decide which of the five squares will turn red (Figure 1). Given that turning red is random, and we have five squares, there is a 20% chance of making correct predictions. Interestingly, participants overestimate their predictive abilities when the time interval between the empty squares and the color change is short [21]. Moreover, those who conspicuously tend to confuse anticipation (an internally generated prediction about which square will be red) and perceptual experience (an external change of color) would also display non-rational higher-level beliefs, often at the level of delusions (e.g., magical thinking, alien control, supernatural powers, thought broadcasting, and future telling). Indeed, Bear et al. (2017) showed that people from the general community with enhanced postdiction scored higher on a scale measuring non-clinical delusional thinking.

There is an enormous amount of evidence that cognitive-behavioral therapy (CBT) is helpful in the treatment of psychotic disorders, including the amelioration of delusions, although the effect size values are only in the mild-to-moderate range [29,30,31]. CBT is regularly added to antipsychotic medications, but it can also be used alone [32,33,34]. However, despite the extensive research on the clinical effectiveness of antipsychotic medications and CBT, fundamental changes in neurocognitive mechanisms during treatment are less known. For example, it has not been explored how antipsychotics and CBT may affect postdiction, which is the very basis of reality testing built on the separation of internal beliefs and external percepts. A possible mechanism is that antipsychotics decrease the salience of predictions (i.e., the confidence in beliefs and predictions) [35], whereas CBT enhances cognitive flexibility and belief updating [36].

The present study aimed to investigate postdiction in schizophrenia and its changes during treatment. We had the following hypotheses: (1) patients with schizophrenia exhibit higher postdiction relative to controls; (2) higher postdiction scores are related to more severe delusions; (3) treatment with CBT and antipsychotics improves both postdiction and clinical symptoms; (4) combined treatment with CBT plus antipsychotics is superior to antipsychotic treatment alone in the normalization of postdiction.

## 2. Materials and Methods

### 2.1. Study Design

In a naturalistic clinical setting (observational study), we assessed two groups of patients with schizophrenia receiving different treatments (antipsychotics and antipsychotics plus CBT) at baseline when approximately the treatment started (week 0) and a follow-up occasion (week 12). We performed the assessments in a quiet room. At baseline and follow-up, patients received a battery of scales and interviews measuring clinical symptoms and completed the postdiction and temporal discrimination tasks. We also included a non-clinical control group to ensure that the tasks were valid for repeated measurements. The postdiction and temporal discrimination tasks were performed in separate runs. The duration of each run was 20–30 min. The author of the study administered the tasks.

### 2.2. Participants

We enrolled 50 patients with schizophrenia at three psychiatric centers in North and South Hungary, coordinated at the National Institute of Psychiatry and Addictions, Budapest, Hungary. The Structured Clinical Interview for DSM-5 Disorders—Clinician Version (SCID-5-CV) confirmed the diagnosis [37]. In addition, all patients scored at least four on the Positive and Negative Syndrome Scale (PANSS) delusions or hallucinations items on the first testing occasion or at least five on suspiciousness, persecution, or grandiosity items [32]. The clinical rating scales and interviews were administered by appropriately qualified raters who were blind to the aim of the study and the treatment status of the patients.

Our observational study was conducted in a natural clinical setting and was not a part of a clinical trial. We followed up two groups: schizophrenia patients on antipsychotics (*n* = 25) and schizophrenia patients on antipsychotics plus CBT (*n* = 25). We tested the patients on two occasions: at week 0 and after a 3-month follow-up period (week 12). At week 0, the patients received the treatment for 5–10 days. Four patients did not complete the treatment phase. We excluded them from the analysis, and they are not reported in the samples defined above.

We enrolled non-clinical control individuals from the community via internet advertisement. The two groups of patients and controls were similar in age, sex distribution, and education (Table 1).

### 2.3. Postdiction Task

We used the postdiction task because, in a previous study, it was associated with delusive thinking [21]. In addition, the task is simple and easy to administer, which is critical in a clinical setting for patients with schizophrenia. Participants observed five empty squares on display. After a time interval, one of the squares turned red. The task was to predict which one of the five squares would turn red [21] (Figure 1). We used a Display++ LCD monitor (Cambridge Research Systems) controlled by a Dell Precision T3640 workstation for stimulus presentation and data collection. We implemented the experiment in a Psychtoolbox3/MATLAB environment (MathWorks).

The first step was that each participant received a detailed written explanation of the task proposed by Bear et al. (2017). Next, the experimenter made sure that they understood the task. Finally, after reading the task description, participants received 20 practice trials to familiarize themselves with the experimental environment.

A trial began by presenting a fixation cross (30-pixel) for 500 ms (Figure 1). Immediately after the fixation cross, five empty squares (50 × 50 pixels) appeared on a 5 × 5 grid (random location, 20-pixel space between each possible square location; the total display area: 330 × 330 pixels). Next, we asked the participants to “pick (in your head) a single square that you think will turn red.” There were two possible time intervals (delays) between the appearance of the five empty squares and the time point when one of these squares turned red: 100 and 1000 ms. The short delay taps on dominantly automatic and fast information processing, whereas the long interval involves attention and working memory [20,21]. We administered 20 trials at each time interval, which varied randomly across the trials.

After one of the squares became red, observers indicated whether they had correctly predicted the color-changing square or not. There were three possible responses linked to different keys on the computer keyboard: “yes, my prediction was correct” (key “i”), “no, I predicted another square” (key “n”), and “I had no time to predict the square” (space bar). The subsequent trial started when the participant pressed the “enter” key following the response. We analyzed two dependent variables: the probability of “yes” responses (the probability of correctly predicting the red square) and the probability of making any prediction considering the ratio of missed trials (“I had no time to predict the square”).

### 2.4. Temporal Discrimination Task

We used a temporal discrimination procedure as a control condition for the prediction task with a similar stimulus set and structure (Bear et al., 2017) (Figure 1). A fixation cross (duration: 500 ms) preceded the five empty squares, which appeared for 500 ms in a trial. Next, two alternative events could happen randomly: the display went blank for 50 ms (all squares disappeared), or one of the squares turned red for 50 ms. The delay period was −100 (red square first) ms or 100 ms (blink first). There were 20 trials at each delay. The task was to decide whether the blink (empty screen) was the first (key “v”) or the red square was the first (key “p”). The dependent variable was the probability of perceiving blink first at each delay (−100 ms—red square first, 0% probability to perceive blink first; 100 ms—blink first, 100% probability of perceiving blink first).

### 2.5. Clinical Assessment

#### 2.5.1. Positive and Negative Syndrome Scale (PANSS)

The PANSS is a clinician-administered, semi-structured interview, rating the patient from 1 (absent) to 7 (severe) on 30 observable and verbally accessible symptoms of schizophrenia and other psychotic disorders. The five-factor model of PANSS incorporates the syndromes into five domains: positive symptoms (delusions, hallucinations, and unusual thought content), negative symptoms (lack of spontaneity and flow of conversation, blunted affect, emotional withdrawal, passive/apathetic social withdrawal, motor retardation, poor rapport, active social avoidance, and uncooperativeness), disorganization (stereotyped thinking, poor attention, disorientation, conceptual disorganization, and difficulty in abstract thinking), excitement (poor impulse control, hyperactivity, hostility, and uncooperativeness), and emotional distress (anxiety, depression, guilt feelings, and tension) [38,39]. The mean kappa reliability coefficient for single items was 0.87 between the two raters.

#### 2.5.2. Peters et al. Delusion Inventory (PDI)

The PDI is a self-rating scale consisting of 21 items [40]. Each item deals with common delusional themes. During the completion of PDI, participants judge whether an item is true or not (e.g., “Do your thoughts ever feel alien to you in some way?”—yes or no; “Do you ever feel as if you are a robot or zombie without a will of your own?”—yes or no). If the item is true, participants are asked to rate how distressing the belief or experience is, how often they think about it (preoccupation), and how true they believe it (conviction) (min: 1, max: 5 points). The maximum total score is 315. The internal consistency (Cronbach’s alpha = 0.85) and the test-retest reliability (*r* = 0.82) of the scale are good.

### 2.6. Cognitive-Behavioral Therapy for Psychosis (CBTp)

During the 12-week treatment period, a trained and experienced therapist delivered CBTp to the patients, following the protocol of the U.S. Department of Veterans Affairs, Mental Illness Research, Education and Clinical Center (MIRECC) [41]. The treatment process included working with hallucinations and delusions, group interventions for delusions and voices, and working with thought disorders and negative symptoms. We regularly supervised the weekly CBTp sessions to ensure fidelity to protocol, and the recorded sessions were rated with the Cognitive Therapy Scale-Revised (CTS-R) [42].

### 2.7. Data Analysis

We used STATISTICA 13.5 (Tibco) for data analysis. Before applying parametric statistical tests, the raw data were entered into Levene’s tests (homogeneity of variance) and Kolmogorov–Smirnov test (normal distribution). As a result, the normality of residuals and homogeneity of variances in the experimental groups met the assumptions.

Separate repeated measures analyses of variance (ANOVAs) were conducted on postdiction measures (probability of predicting the red square and probability of making predictions) and temporal discrimination measures (probability of correct responses). In both ANOVAs, the experimental group (antipsychotics, antipsychotics plus CBT, and control subjects) was the between-subjects factor. The within-subjects factors were treatment (0 vs. 12 weeks) and delay (100 vs. 1000 ms) in the postdiction task. In the temporal discrimination task, the within-subjects factor was only the treatment because the two conditions (red square first and blink first) were treated separately, and the dependent variable was the probability of correct responses. Tukey’s Honestly Significant Differences (HSD) tests were applied for post-hoc comparisons.

Multiple regression analyses investigated the relationship between postdiction and temporal discrimination scores and clinical measures. The potential predictors of postdiction and temporal discrimination tasks were: PANSS positive, negative, disorganized, excitement, and emotional discomfort scores, PDI conviction, preoccupation, and distress scores. Sex (categorical predictor), age, and education were also considered as potential confounding factors in the regression analyses.

The level of statistical significance (the probability of type I errors) was set at *p* < 0.05. We did not use corrections for multiple comparisons.

## 3. Results

### 3.1. Postdiction Performance

We focused on postdiction scores at short (100 ms) and long (1000 ms) delays at 0 and 12 weeks in the two treatment groups (antipsychotics and antipsychotics plus CBT) and non-clinical control individuals. The results from the ANOVA are summarized in Table 2.

As shown in Figure 2, at week 0, patients with schizophrenia exhibited higher postdiction scores relative to non-clinical control individuals at 100 ms delay, but the two schizophrenia groups did not differ (Tukey’s test, controls vs. antipsychotics: *p* = 0.04; controls vs. antipsychotics + CBT: *p* = 0.003; antipsychotics vs. antipsychotics + CBT: *p* = 0.9). At week 12, no significant difference persisted between patients, including those who received antipsychotics alone and antipsychotics combined with CBT, and controls (Tukey’s test, controls vs. antipsychotics: *p* = 0.47; controls vs. antipsychotics + CBT: *p* = 0.47; antipsychotics vs. antipsychotics + CBT: *p* = 0.45).

A within-group comparison contrasting short delay (100 ms) postdiction performances at 0 and 12 weeks indicated a significant decrease only in the antipsychotics plus CBT group (*p* = 0.0001) but not in the antipsychotics alone group (*p* = 0.86). Finally, at the long delay (1000 ms), we observed similar performances in patients and controls at both assessments (all *p*s > 0.5) (Figure 2).

### 3.2. Probability of Making Predictions in the Postdiction Task

Figure 3 displays the results, and Table 3 summarizes the outcome from the ANOVA. Except for the main effect of delay, the ANOVA revealed no significant main effects and interactions.

### 3.3. Temporal Discrimination Performance

Figure 4 and Table 4 depict the results and the ANOVA outcomes. Tukey’s tests revealed that the patients with schizophrenia exhibited worse temporal discrimination performances than the controls at week 0 (controls vs. antipsychotics: *p* = 0.0001; controls vs. antipsychotics + CBT: *p* = 0.002; antipsychotics vs. antipsychotics + CBT: *p* = 0.9) and week 12 (controls vs. antipsychotics: *p* = 0.001; controls vs. antipsychotics + CBT: *p* = 0.01; antipsychotics vs. antipsychotics + CBT: *p* = 0.9).

### 3.4. Test-Retest Reliability of Postdiction and Temporal Discrimination

As shown in Figure 2, Figure 3 and Figure 4, the control individuals exhibited very similar performances on all measures when the first (week 0) and second (week 12) sessions were compared (all *p*s > 0.5). The correlations between postdiction and temporal discrimination measures at weeks 0 and 12 were high (all *r*s > 0.8).

### 3.5. Clinical Outcomes

#### 3.5.1. PANSS

The PANSS scores and their comparisons are depicted in Table 5 and Table 6. There were no significant main effects of the group for all measures, which indicates similar scores in schizophrenia patients receiving antipsychotics and those on antipsychotics plus CBT. However, the effect of treatment was significant on each PANSS score. In the case of positive symptoms and emotional discomfort, we found a two-way interaction between group and treatment, indicating a more pronounced treatment effect in the antipsychotics plus CBT group than in the antipsychotics group (Table 5 and Table 6).

#### 3.5.2. PDI

Like on the PANSS, we obtained no significant main effects of group (antipsychotics vs. antipsychotics plus CBT), but the treatment effect was significant. Additionally, there were significant interactions between group and treatment, suggesting a higher decrease in mean PDI scores in the antipsychotics plus CBT group than in the antipsychotics group (Table 5 and Table 6).

### 3.6. Relationship between Postdiction and Clinical Measures

At 0 week, PANSS positive, PANSS emotional distress, and PDI conviction values predicted postdiction scores at short delay when corrected for age, education, and sex (PANSS positive: *b** = 0.58, *SE* = 0.12, *p* < 0.001; PANSS emotional discomfort: *b** = 0.36, *SE* = 0.13, *p* = 0.008; PDI conviction: *b** = 0.34, *SE* = 0.14, *p* = 0.02). However, only the PANSS positive (*r* = 0.59, *p* < 0.001) and PDI conviction scores (*r* = 0.33, *p* = 0.02) correlated with postdiction. The PANSS emotional discomfort scores–postdiction correlation was not significant (*r* = 0.21, *p* = 0.15) (Figure 5).

At 12 weeks, these relationships did not retain statistical significance (PANSS positive: *b** = 0.32, *SE* = 0.16, *p* = 0.05; PANSS distress: *b** = 0.08, *SE* = 0.15, *p* = 0.60; PDI conviction: *b** = 0.13, *SE* = 0.14, *p* = 0.40). There was no significant relationship between the clinical measures and postdiction at long delay (*p*s > 0.5).

We also investigated whether changes in postdiction scores during the treatment were related to changes in clinical symptoms during the follow-up period (0 vs. 12 weeks). We did not prove that changes in postdiction scores were predicted by improvements in clinical symptoms (postdiction changes—PANSS positive changes: *r* = 0.15, *p* = 0.32; postdiction changes—PANSS emotional distress changes: *r* = 0.12, *p* = 0.40; postdiction changes—PDI conviction changes: *r* = 0.27, *p* = 0.06).

## 4. Discussion

In line with our hypothesis, patients with schizophrenia showed enhanced postdiction at the baseline assessment, which correlated with the positive symptoms, delusional conviction, and emotional distress associated with psychosis. The main finding was that the antipsychotic medication plus CBT combination markedly reduced postdiction during the three-month treatment phase. In addition, the treatment did not affect temporal discrimination performance, indicating a specific impact on postdiction, which is a putative cognitive marker of delusions. The reduction of postdiction during treatment indicates that patients with schizophrenia improved in reality testing, namely, in the discrimination of internal thoughts (predictions on future color changes of squares) and perceptual changes in the external world (physical changes of color). The results confirm that thought–percept separation is a fundamental aspect of reality testing, which is substantially ameliorated during combined antipsychotic and CBT treatment. Furthermore, the degree of improvement in postdiction was more significant in the antipsychotics plus CBT condition than that observed when only antipsychotics were applied.

It is of particular importance how antipsychotics affect postdiction. Increased dopamine synthesis in the ventral striatum is a critical mechanism for strengthening erroneous predictions in schizophrenia, that is, abnormal salience attribution to neutral stimuli and an enhanced conviction in wrong predictions [43,44,45,46]. We speculate that elevated striatal dopamine levels may also contribute to abnormal postdiction in schizophrenia. By blocking dopamine receptors, antipsychotics dampen down salience attribution to predictions and might reduce postdiction by separating predictive thoughts and percepts [35]. CBT may add extra value to the treatment process via different mechanisms. CBT modulates selective attention, executive functions, attributions, and learning and memory by strengthening the adaptive and flexible functioning of the prefrontal cortex [36,47]. Altogether, the antipsychotics plus CBT combination targets symptoms and cognitive biases focusing on two distinct levels: bottom-up reduction of abnormal salience to predictions (less conviction in beliefs) by antipsychotics and the enhancement of flexible top-down control from the prefrontal cortex (revision and updating of beliefs).

The present study’s findings resonate with a previous report revealing that subclinical delusive thinking in the general population was associated with postdiction at short thought–percept time intervals [21]. When the delay between predictive thought and perceptual changes is short, information processing is more automatic than at more extended time intervals when participants store their predictions in working memory before the color changes. Therefore, the comparison of predictive thought and perceptual changes happens in a controlled manner at long time intervals. Even though schizophrenia and schizotypal individuals generally show working memory impairments [48,49], we did not find differences between patients and controls in postdiction performance at long time intervals. Similarly, patients with schizophrenia exhibited less punctual temporal discrimination of stimulus sequences than the control individuals, but this cognitive deficit was stable during the treatment. This finding is consistent with a large body of literature showing that patients with schizophrenia perform poorly on tasks requiring the judgment of time intervals and temporal order of events [50].

Some limitations of the present study must be mentioned. First, the study was conducted in a natural clinical setting and did not meet the rigor of randomized-controlled clinical trials. However, the degree of clinical improvement during the follow-up period was comparable to that reported in previous clinical trials [32]. Furthermore, the two schizophrenia groups (antipsychotics and antipsychotics plus CBT) were similar in demographic and clinical parameters at the first assessment. Therefore, potential differences in demographic features and clinical symptoms do not explain the distinct degree of improvement in postdiction in the two treatment groups. Second, we markedly shortened the postdiction and temporal discrimination tasks: we used only two delays. This modification of the experimental procedure was indispensable to make the task easier for the patients and avoid fatigue.

Nevertheless, we could investigate the time intervals in which the most significant relationship was found between task performance and delusive thinking in previous studies [21]. We also had appropriate control conditions (long time intervals in the postdiction and temporal discrimination tasks). Finally, the sample size was small, although comparable to previous clinical trials [32]. We did not prove that reduction in postdiction was associated with the improvement of clinical symptoms, possibly because the sample size was insufficient to detect such correlations.

## 5. Conclusions

The most important conclusion of this study is that postdiction may serve as a marker of reality testing (i.e., distinguishing between thought and perception) in clinical settings, which correlates with delusions and exhibits a robust change during treatment. Given that the main aim of CBT in psychosis is to enhance reality testing by specifically targeting abnormal attributional processes associated with delusive thinking, it is apparent that postdiction exhibited normalization together with the improvement of clinical symptoms. One promising possibility is that postdiction can be extended to other functions, including sensory-motor coordination, memory, and higher-level cognition, serving as a “free will” model and the feeling of agency [23].

## Figures and Tables

**Figure 1 behavsci-12-00198-f001:**
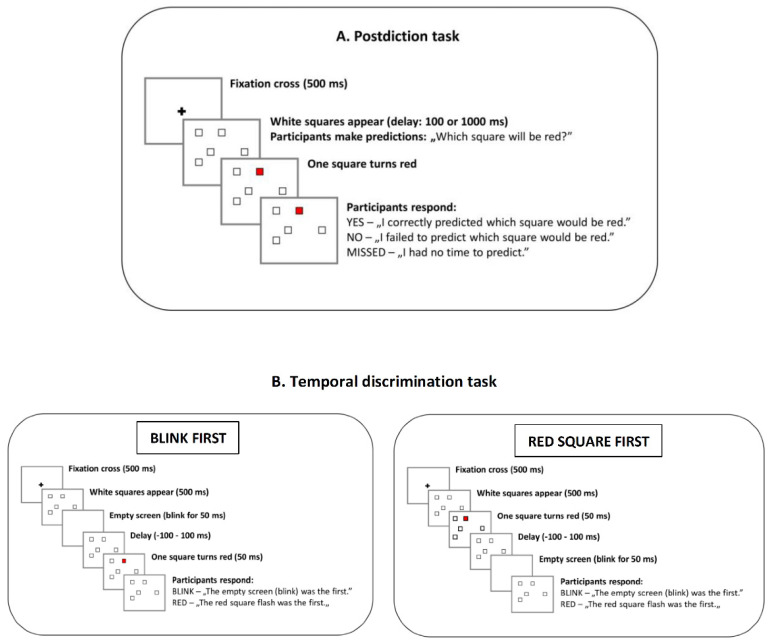
*Postdiction task* (**A**). Five empty squares were presented on the display. Participants predicted which of the five squares would turn red. *Temporal discrimination task* (**B**). After presenting the five squares, the screen blinked (blink first), or one of the squares turned red (red square first). Participants indicated whether they observed the blink or the flashed red square first. Following the literature convention, negative delay (−100 ms) refers to the case when the red square is the first.

**Figure 2 behavsci-12-00198-f002:**
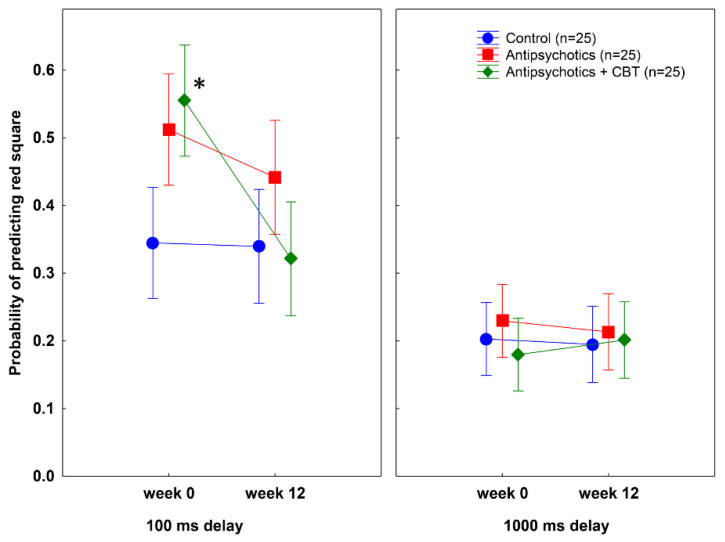
Mean probabilities of predicting the red square at each delay (100 ms and 1000 ms) at weeks 0 and 12. Error bars indicate 95% confidence intervals. At week 0, patients with schizophrenia achieved higher postdiction scores than controls when the delay was short (100 ms) (* antipsychotics vs. control: *p* = 0.04; antipsychotics plus CBT vs. control: *p* = 0.003, Tukey’s test) but not when it was long (1000 ms). At week 12, there was no significant difference between patients and controls. A significant drop in postdiction during the 12-week treatment phase was observed only in patients who received combined antipsychotic treatment and cognitive-behavioral therapy (CBT) (week 0 vs. 12, 100 ms delay, *p* = 0.0001).

**Figure 3 behavsci-12-00198-f003:**
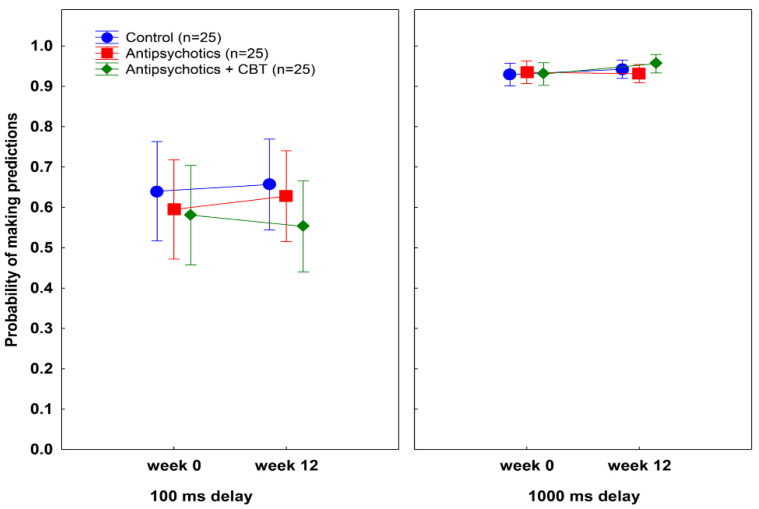
Mean probabilities of making predictions at each delay (100 ms and 1000 ms) at weeks 0 and 12. Error bars indicate 95% confidence intervals. There were no significant differences among the groups.

**Figure 4 behavsci-12-00198-f004:**
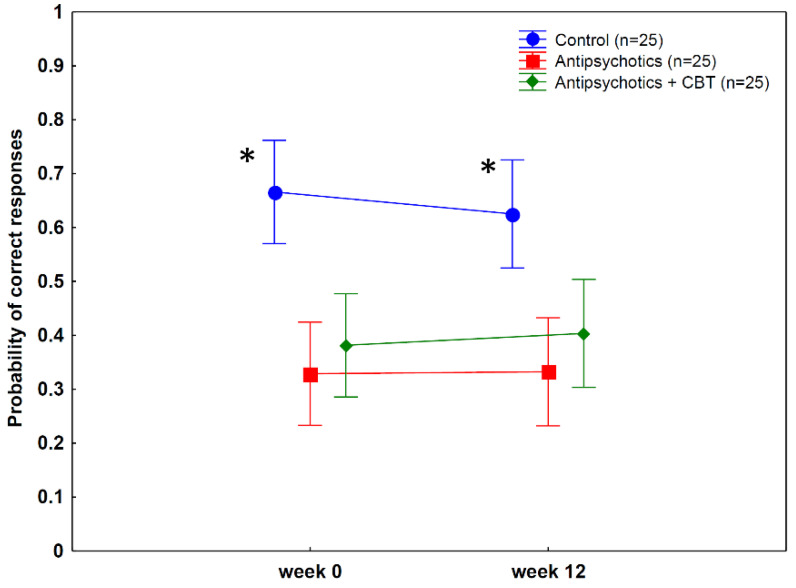
Temporal discrimination performance (mean probability of correct responses, 95% confidence intervals). Patients with schizophrenia were less accurate than controls both before and after treatment. (* week 0, controls vs. antipsychotics: *p* = 0.0001; controls vs. antipsychotics plus CBT: *p* = 0.002; week 12, controls vs. antipsychotics: *p* = 0.001; controls vs. antipsychotics + CBT: *p* = 0.01, Tukey’s test).

**Figure 5 behavsci-12-00198-f005:**
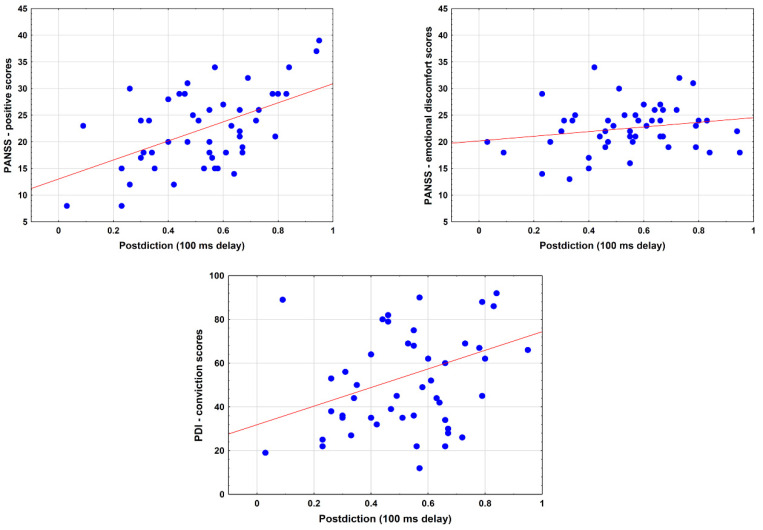
Correlation between postdiction at short delay (100) ms and clinical measures (PANSS—Positive and Negative Syndrome Scale, PDI—Peters et al. Delusion Inventory).

**Table 1 behavsci-12-00198-t001:** Demographic and clinical characteristics of the participants.

	Controls (*n* = 25)	Antipsychotics (*n* = 25)	Antipsychotics Plus CBT (*n* = 25)
Male/female	15/10	17/8	18/7
Age (years)	38.8 (12.9)	39.8 (10.9)	35.9 (9.2)
Education (years)	11.4 (2.3)	11.6 (2.7)	11.5 (2.5)
Duration of illness (years)	-	8.7 (2.6)	9.3 (2.8)
Type of antipsychotics	-	Olanzapine (*n* = 11) Amisulpride (*n* = 7) Clozapine (*n* = 6)	Olanzapine (*n* = 13) Amisulpride (*n* = 5) Clozapine (*n* = 6)

Data are mean (standard deviation) for age, education, and duration of illness.

**Table 2 behavsci-12-00198-t002:** Summary of the results from the ANOVA conducted on postdiction scores (probability of predicting red square).

	*df*	*F*	*p*	*η* ^2^
Main effect of group (antipsychotics, antipsychotics + CBT, non-clinical controls)	2, 72	3.47	0.04	0.09
Main effect of treatment (week 0 vs. 12)	1, 72	12.91	0.001	0.15
Main effect of delay (100 vs. 1000 ms)	1, 72	80.25	<0.0001	0.53
Group by treatment interaction	2, 72	4.02	0.02	0.10
Group by delay interaction	2, 72	2.25	0.11	0.06
Treatment by delay interaction	1, 72	9.12	0.004	0.11
Group by treatment by delay interaction	2, 72	5.36	0.007	0.13

**Table 3 behavsci-12-00198-t003:** Summary of the results from the ANOVA conducted on postdiction scores (probability of making predictions).

	*df*	*F*	*p*	*η* ^2^
Main effect of group (antipsychotics, antipsychotics + CBT, non-clinical controls)	2, 72	0.79	0.46	0.02
Main effect of treatment (week 0 vs. 12)	1, 72	0.14	0.71	0.001
Main effect of delay (100 vs. 1000 ms)	1, 72	181.44	<0.0001	0.72
Group by treatment interaction	2, 72	0.04	0.96	0.001
Group by delay interaction	2, 72	1.13	0.33	0.03
Treatment by delay interaction	1, 72	0.01	0.93	0.0001
Group by treatment by delay interaction	2, 72	0.27	0.76	0.008

**Table 4 behavsci-12-00198-t004:** Summary of the results from the ANOVA conducted on the temporal discrimination performance (probability of correct answers).

	*df*	*F*	*p*	*η* ^2^
Main effect of group (antipsychotics, antipsychotics + CBT, non-clinical controls)	2, 2	12.03	0.00003	0.25
Main effect of treatment (week 0 vs. 12)	1, 72	0.19	0.67	0.003
Group by treatment interaction	2, 72	2.54	0.09	0.07

**Table 5 behavsci-12-00198-t005:** Summary of ANOVAs conducted on the clinical scales in the schizophrenia groups treated with antipsychotics and antipsychotics plus CBT.

	*df*	*F*	*p*	*η* ^2^
**PANSS—Positive**				
Group	1, 48	0.004	0.95	0
Treatment	1, 48	89.80	<0.0001	0.65
Group by treatment	1, 48	18.48	0.0001	0.28
**PANSS—Negative**				
Group	1, 48	1.43	0.24	0.03
Treatment	1, 48	27.05	<0.0001	0.36
Group by treatment	1, 48	0.72	0.40	0.02
**PANSS—Disorganized**				
Group	1, 48	1.21	0.28	0.02
Treatment	1, 48	65.34	<0.0001	0.58
Group by treatment	1, 48	1.68	0.20	0.03
**PANSS—Excitement**				
Group	1, 48	0.24	0.62	0.01
Treatment	1, 48	85.65	<0.0001	0.64
Group by treatment	1, 48	1.91	0.17	0.04
**PANSS—Emotional discomfort**				
Group	1, 48	0.003	0.96	0
Treatment	1, 48	105.91	<0.0001	0.69
Group by treatment	1, 48	9.35	0.004	0.16
**PDI—Conviction**				
Group	1, 48	0.05	0.83	0.001
Treatment	1, 48	21.38	<0.0001	0.31
Group by treatment	1, 48	4.87	0.03	0.09
**PDI—Preoccupation**				
Group	1, 48	0.23	0.63	0.005
Treatment	1, 48	106.87	<0.0001	0.69
Group by treatment	1, 48	21.66	<0.0001	0.31
**PDI—Distress**				
Group	1, 48	0.46	0.50	0.009
Treatment	1, 48	65.0	<0.0001	0.58
Group by treatment	1, 48	4.39	0.04	0.08

PANSS—Positive and Negative Syndrome Scale; PDI—Peters et al. Delusion Inventory; CBT—Cognitive-Behavioral Therapy.

**Table 6 behavsci-12-00198-t006:** Scores on the clinical scales and their comparisons in the schizophrenia groups treated with antipsychotics and antipsychotics plus CBT.

	0 Week		12 Weeks		0 vs. 12 Weeks	
	Antipsychotics (*n* = 25)	Antipsychotics + CBT (*n* = 25)	Antipsychotics (*n* = 25)	Antipsychotics + CBT (*n* = 25)	Antipsychotics	Antipsychotics + CBT
PANSS—Positive	21.5 (7.5)	23.6 (6.8)	19.2 (7.7)	17.3 (6.1)	*d* = 0.30 *p* = 0.004	*d* = 0.98 *p* < 0.001
PANSS—Negative	15.5 (4.1)	16.4 (4.3)	13.2 (3.9)	14.8 (3.3)	*d* = 0.58 *p* = 0.001	*d* = 0.45 *p* < 0.05
PANSS—Disorganized	13.0 (2.0)	12.7 (2.1)	11.6 (1.8)	10.8 (2.1)	*d* = 0.73 *p <* 0.001	*d* = 0.90 *p <* 0.001
PANSS—Excitement	15.8 (3.1)	16.0 (3.6)	12.7 (2.6)	11.8 (2.9)	*d* = 1.09 *p <* 0.001	*d* = 1.29 *p <* 0.001
PANSS—Emotional discomfort	21.8 (4.1)	23.2 (4.6)	18.4 (3.8)	17.0 (4.4)	*d* = 0.86 *p <* 0.001	*d* = 1.38 *p <* 0.001
PDI—Conviction	50.5 (26.8)	58.5 (26.3)	43.0 (19.7)	37.3 (18.7)	*d* = 0.31 *p <* 0.05	*d* = 0.94 *p <* 0.001
PDI—Preoccupation	43.7 (20.0)	47.8 (19.9)	35.9 (15.5)	27.1 (16.3)	*d* = 0.44 *p* = 0.001	*d* = 1.14 *p <* 0.001
PDI—Distress	43.5 (28.7)	42.3 (26.1)	34.0 (23.2)	22.6 (17.8)	*d* = 0.37 *p* = 0.001	*d* = 0.90 *p <* 0.001

Data are mean (standard deviation). The *p*-values show comparisons with Tukey’s tests. The *d*-values are Cohen’s effect size. Both *p-* and *d*-values indicate comparisons between values at weeks 0 and 12. PANSS—Positive and Negative Syndrome Scale, PDI—Peter’s et al. Delusion Inventory, CBT—Cognitive-Behavioral Therapy.

## Data Availability

Raw data are available in the Appendix A.

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
