# Peer review of "Postdiction in Visual Awareness in Schizophrenia"

_behavsci, 2022, doi:10.3390/bs12060198_

Round 1

Reviewer 1 Report

This is a study investigating the effect of antipsychotics and the combination of antipsychotics and cognitive-behavioral therapy on postdiction in visual awareness in people with schizophrenia. The paper is well-written and interesting for the journal and the readers; however, several minor changes should be made before publishing it.

The abstract section is well written. I would add some more words about the method. How were the groups divided? It is an observational study. I consider it should be clarified that this is not a randomized trial. What was the main aim of the study? Did the authors describe a hypothesis?

The introduction is really good. I would also add more recent references about the formation of delusions according to the different theories: cognitive... etc.

How is the hypothesis about the modification of postdiction in visual awareness by antipsychotics? In the introduction, the authors have explained evidence about CBT but the effectiveness of antipsychotics have been poorly explained. I would recommend to add two references.

In the methods section the authors are starting with a subsection of participants. I would prefer to better start with study design.

The subsection about the PANSS, PDI, CBTp should be grouped in a subsection called (for example): psychometric assessment.

The discussion section is brief. I would add  two more paragraphs adding discussion to the effect of antipsychotics (and which ones) and CBT on postdiction.

Author Response

We thank the insightful comments and suggestions of the reviewer. Below please find the responses (R1) to each point (P) raised. The modifications in the manuscript are highlighted in red.  

P1: "The abstract section is well written. I would add some more words about the method. How were the groups divided? It is an observational study. I consider it should be clarified that this is not a randomized trial. What was the main aim of the study? Did the authors describe a hypothesis?"

R1: We added the fact that the division of the groups was the decision of the clinician (no randomization) (line 18). We also stressed that it was not a randomized trial (line 16). We added the hypothesis to the aim of the study (line L 14-15). 

P2: "The introduction is really good. I would also add more recent references about the formation of delusions according to the different theories: cognitive... etc."

R2: New references are added to the introduction including classics and some from the past few years (L 54).

P3: "How is the hypothesis about the modification of postdiction in visual awareness by antipsychotics? In the introduction, the authors have explained evidence about CBT but the effectiveness of antipsychotics have been poorly explained. I would recommend to add two references."

R3: This is a particularly important point. Even in the introduction we refer to this problem as requested, and, in accordance with another request of the reviewer (L 105-107), the mechanism of antipsychotics vs. CBT is discussed in a separate paragraph at the end of the paper (L 499-513). 

P 4-5: "In the methods section the authors are starting with a subsection of participants. I would prefer to better start with study design. The subsection about the PANSS, PDI, CBTp should be grouped in a subsection called (for example): psychometric assessment."

R4-5: These modification in the structure of the manuscript have been implemented in the revision (L 127-137; L 212). 

P6: "The discussion section is brief. I would add  two more paragraphs adding discussion to the effect of antipsychotics (and which ones) and CBT on postdiction."

R6: The new section has been added to the discussion with new references (L 499-513). 

Reviewer 2 Report

The paper reports a well-designed study on postdiction in schizophrenia, that may bring an important contribution to the research on the cognitive impairments in schizophrenia. I wish to compliment the author for this work. I only have a minor comment, i.e., a short overview of the cognitive associations of schizophrenia included in the Introduction, especially of those relevant for postdiction, would, in my view, provide the reader with a more fit conceptual preparedness to appreciate the relevance of the results, and overall the merit of the paper.

Author Response

We thank the expert review of our manuscript. We are happy that the reviewer expressed a positive opinion in relation to our work. 

The requested short introduction on cognitive associations with schizophrenia is included in the revised manuscript (L 44-50). 

Reviewer 3 Report

I enjoyed reading the author’s manuscript for both its content and its scientific soundness. It reports a very interesting experiment on schizophrenic patients which is, in my opinion, very useful for the current scientific community dealing with psychotic traits in both healthy and clinical populations.

Although my overall judgement of the manuscript is very good, I think that the manuscript must be thoroughly reviewed before its publication since – in its current form – it is not reproducible and describes some methods/results in an inconvenient and biased way. Also, the introduction should also be revised to better incorporate the agency dimension (which is relevant but only timidly treated in the manuscript).

1)   Introduction

The author efficiently describes the postdiction process and its alterations. I agree with the view that thoughts on expectations and predictions can be confused with percepts and this mistiming may concur in the generation of delusions. Moreover, a relevant part of the text, which I particularly appreciated, is dedicated to the link of postdiction and perception independently of psychotic illnesses.

The presence of such mistiming/misperceptions in non-clinical populations is now extensively known and has been studied well in the context of implicit measures/mechanisms of agency. I encourage the author to dig deeper into these studies and to report the associations of his study with the sense of agency and, most of all, with recent studies reporting the associations between agency and delusions.

See:

- Di Plinio, S, Arnò, S, Perrucci, MG, Ebisch, SJH (2020). The evolving sense of agency: Context recency and quality modulate the interaction between prospective and retrospective processes. Consciousness and Cognition, 80(102903), 1-12, DOI: 10.1016/j.concog.2020.102903

- Moore, JW, Haggard, P (2008). Awareness of action: Inference and prediction. Consciousness and Cognition. 17(1), 136–144. https://doi.org/10.1016/j.concog.2006.12.004

2)   Methods

-      Methods. 2.2:
Please add a more detailed rationale behind the task selection: why were these two tasks chosen? Which are the main strengths of these tasks?
L143-144: Please define here the rationale behind using two delays.
L151-152: Just an observation of mine: At the perceptual level, it can be hypothesized that foveal stimuli and peripheral stimuli are differently valued, that would be true also for individuals with abnormal psychopyshiological decoding of the world. Did the author investigate if the closeness to the fixation cross (of the red square) impacted measured variables? If true, this may open interesting considerations.

-      Methods. 2.3
Please clearly state if the two tasks were performed in separate runs or not. Please state the average trial length and the average time employed by participants to perform each run. Please also report relevant psychometric information: where was the tasks administered, by whom, how where the participants instructed, …?

-      Please note that the negative delay does not make sense. This task has two experimental conditions: flash first or red first. Please change the whole manuscript accordingly.

-      Relatedly, figure 1 should be modified. First, please more accurately describe the Temporal Discrimination task: From the Figure, a reader may not immediately grasp that there are two alternatives (red first vs flash first). Please correct adding one example for each experimental condition. Second, please add subfigures references (A, B). Please note that this second point is valid for all the figures in the mauscript.

-      Methods. 2.7. The data analysis section does not describe appropriately the analysis performed, the contrasts of interest, and the rationale behind the choice of the analysis.

o   Please report analyses on the two tasks separately since they have not the same structure

o   - Controls???

o   - Please accurately describe the regression models! Describe which were the continuous, categoric, and eventually random intercepts/slopes.

o   - Please describe if model assumptions were met (normality of residuals, homoschedasticity).

o   - Tests of interest? Please describe

o   - What does “alpha” mean here?

3)   Results

-      General (and very crucial) observation: It seems to me that data was analyzed as follows:

o   ANOVA: “DV = group * experimental condition * treatment”

§  In the case of the Postdiction task, DV=% of squares predicted; experimental condition=[delay100, delay1000]

§  In the case of the Temporal Discrimination Task experimental condition=[red first, flash first]. However, I can’t really make sense of what the dependent variable (DV) here is! Please note that if the DV is “probability of perceiving blink first”, then the analysis is biased and wrong since the DV in the experimental condition “red first” will be equal to 1 – the DV in the experimental condition “flash first”, and viceversa. This makes the analyses of this task not trustable. Please use “% of correct trials” and treat “red first” and “flash first” as two experimental conditions.

o   Regressions: DV = ???

o   I hypothesized “DV =  group * experimental condition * treatment * PANSS” but this is unlikely. Perhaps a separate regression was run for each group?

o   In any case, the regression suffers the same limitation described above for the DV in the temporal discrimination task. Please revise and re-analyze data accordingly. Moreover, accurately describe the regression model(s) to make your paper reproducible and the results trustable.

-      Report exact p values instead of ranges. Also, *report effect sizes* for each effect of interest. Also state if multiple comparison corrections were performed or not (which are both acceptable).

-      Insert tables with effect sizes and significance to increase readability.

-      I can see that df were corrected. How were degrees of freedom calculated? Please explain all the corrections in the Methods section.

-      L225: “heightened postdiction performances”: Does the author think that this is this a meaningful way to label variables? There is not a real "performance" here. Instead, patients seem to have increased postdiction *scores*

-      L226: HSD Test of what? Please be more specific.

-      In each Figure: What are the asterisks indicating?

-      L259: I am quite convinced that the contrast “control vs schizophrenics (pooled)” should be significant, and I wonder why this was not tested since it is informative (however, there still is the problem of the DV [see above]).

-      3.4: How is this a test-retest? I am a bit confused here.

-      3.6: Please represent significant associations with PANSS scales. Please also directly test associations at week 0 vs week 12. Otherwise, observations of inferences about how the treatment has influenced the association cannot be made. Please also note that in order to solve this issue the author must deal with my first observation above related to the models.

4)   Other

L119: “the” here may be removed for improved readability.

Author Response

We are indebted for the detailed review of our paper and for the insightful comments.  The revised text is depicted in red. The responses and the list of revisions are as follows:

  1. Introduction: We included the requested literature on agency in a new paragraph (L 74-83).
  2. Methods 2.2.: The rationale of task selection and short/long delay are included in the revision (L 169-171). The idea that peripheral and central location of stimuli may impact processing is extremely interesting and merits future studies. However, in the present study, we did not investigate this issue.  
  3. Methods 2.3.: The details of trials (length, separate run, administration) are now included in a separate paragraph on general study design as requested by another reviewer (L 128-138). 
  4. Figure 1 have been fully modified to illustrate the two different conditions in the temporal discrimination task (blink first and red square first) (L 115-116). We agree that the negative delay seems to be odd, but we did not change it to retain a similar terminology to other studies in the literature (Bear et al., 2016, 2017). Moreover, negative delay is not entirely meaningless and nonsense. Negative delay is used in the red square first condition in the temporal discrimination task. Indeed, in this case, the delay is inserted after the target stimulus (red square) in contrast to the case when the delay is really before the target stimulus (blink first condition). In classic psychophysical paradigms, delay is usually before the target stimulus, which is defined as positive (real) delay. 
  5. Methods, data analysis section: This part has been rewritten to meet the suggestions. The regression model and the model assumption are included in the revision. Please note that the postdiction and temporal discrimination tasks had similar structures, and, therefore, the ANOVAs were also similar for both tasks. Alpha refers to the probability of type I errors (p) (L 257-267). 
  6. Results: This section has been fully rewritten according to the in-depth criticism and suggestions. Below please find the list of key changes:
  • The analysis of the temporal discrimination task has been modified (L 344-358).
  • Exact p-values and effect size are summarized in Tables (new Tables 2-5). 
  • The abbreviation HSD is defined in the methods section (L 260-261). 
  • Test-retest analysis is described in more details (L 377-380). 
  • Correlations with PANSS scores are shown, and the 0 vs. 12 weeks are compared (new Fig. 5, L 471-474; L 478-481). 
  • Original L119 (now 157): "the" is eliminated. 

Round 2

Reviewer 3 Report

The author made a consistent and solid work in correcting and updating the current manuscript. Thanks. I am happy with the changes made.

As an answer to the author's point "Negative delay is used in the red square first condition in the temporal discrimination task. Indeed, in this case, the delay is inserted after the target stimulus (red square) in contrast to the case when the delay is really before the target stimulus (blink first condition). In classic psychophysical paradigms, delay is usually before the target stimulus, which is defined as positive (real) delay. " Please note that "negative delay" is often a misleading definition since it automatically suppose a symmetry with the "positive delay", but these are -often- more logically labelled as two separate experimental conditions. However, this discussion is not relevant and the author has suited the manuscript to match both views. 

Please correct the points below before publication:

L268: Define "alpha". Does it refer to p-values after multiple comparisons? Which correction was made?

L306: Report exact p value.

L357: Report exact p value.

L474: Report each test made with exact p value and r value.

Author Response

L268: Define "alpha". Does it refer to p-values after multiple comparisons? Which correction was made?"

R: Alpha is basically the p-value. We did not use corrections for multiple comparisons. The main reason is that Tukey's post hoc tests are very conservative, and here corrections are not needed.  We made it clear in the paper. 

"L306: Report exact p value."

R: Done. 

"L357: Report exact p value."

R: Done. 

"L474: Report each test made with exact p value and r value."

R: Done.